# Viral Vector-Based Gene Therapy

**DOI:** 10.3390/ijms24097736

**Published:** 2023-04-23

**Authors:** Xuedan Li, Yang Le, Zhegang Zhang, Xuanxuan Nian, Bo Liu, Xiaoming Yang

**Affiliations:** 1National Engineering Technology Research Center for Combined Vaccines, Wuhan 430207, China; xuedanli@whu.edu.cn (X.L.); 13135687868@163.com (Y.L.); zhangzhegang@sinopharm.com (Z.Z.); nianxuanxuan@126.com (X.N.); liubohust@126.com (B.L.); 2Wuhan Institute of Biological Products Co., Ltd., Wuhan 430207, China; 3China National Biotech Group Company Limited, Beijing 100029, China

**Keywords:** gene therapy, adeno-associated viruses, adenoviruses, lentiviruses, challenges

## Abstract

Gene therapy is a technique involving the modification of an individual’s genes for treating a particular disease. The key to effective gene therapy is an efficient carrier delivery system. Viral vectors that have been artificially modified to lose their pathogenicity are used widely as a delivery system, with the key advantages of their natural high transduction efficiency and stable expression. With decades of development, viral vector-based gene therapies have achieved promising clinical outcomes. Currently, the three key vector strategies are based on adeno-associated viruses, adenoviruses, and lentiviruses. However, certain challenges, such as immunotoxicity and “off-target”, continue to exist. In the present review, the above three viral vectors are discussed along with their respective therapeutic applications. In addition, the major translational challenges encountered in viral vector-based gene therapies are summarized, and the possible strategies to address these challenges are also discussed.

## 1. Introduction

Human gene therapy seeks to modify or manipulate the expression of a gene or alter the biological properties of living cells for a therapeutic objective. This technology has the potential to cure diseases that are treatable and not entirely curable with conventional medications, thereby providing treatments for diseases previously classified as untreatable [1]. Therefore, human gene therapy is one of the research hotspots in the current century [2]. Recently, several gene therapies have been approved by regulators across the world for the treatment of various conditions, including cancer, blindness, and metabolic disorders [3]. For instance, Zolgensma (onasemnogene abeparvovec) has been approved as a treatment for spinal muscular atrophy, while Luxturna (voretigene neparvovec) has been approved for treating a form of retinal dystrophy capable of causing blindness. As early as 1970, Stanfield Rogers, an American doctor, attempted to treat argininemia by injecting a papillomavirus containing arginase. This was the first human gene therapy trial and is, therefore, an important event in scientific research history, even though it ended in failure [4]. In 1990, William French Anderson et al. conducted a trial for the correction of adenosine deaminase (ADA) deficiency by injecting the T cells transformed using a recombinant retrovirus carrying the ADA gene. This was the first gene therapy protocol that was federally approved [5]. Since then, it has been a long journey (Figure 1), with gene therapy currently producing novel treatment options in multiple fields of medicine. The current clinical trials and approved drugs for gene editing are listed in Table 1 and Table 2, respectively.

In comparison to traditional treatment options, such as proteins or small molecules, which may require repeated infusion, gene therapy delivered to long-lived cells could provide sustained production of endogenous proteins [8], i.e., gene therapy offers “one-shot” curative benefits following the introduction of correct genetic material into the patients in principle [9]. Moreover, no “impossible target medicine” situation exists, and first-time potentially curative options would be offered to all patients [10].

Four basic gene therapy approaches exist, which are as follows: gene supplementation [11,12], gene silencing [13], gene replacement [14], and gene editing [15]. The initial gene therapy focused on the delivery of transgenes, i.e., the repairment or replacement of defective genes [16]. Later, with the rapid development of functional genomics and the iterative diversification of nucleases related to gene editing, an unprecedented revolution occurred in gene therapy approaches. The advent of clustered regularly-interspaced short palindromic repeats/CRISPR-associated protein 9 (CRISPR/Cas9) technology has rendered gene editing much simpler and effective and has, therefore, led to a breakthrough in the field of gene editing after meganucleases, the first generation of gene editing technology based on the zinc finger nucleases (ZFNs) technology, and the second generation of gene editing technology based on the transcription activation-like effector nuclease (TALEN) technology [17,18]. In particular, engineering the Chimeric Antigen Receptor-T (CAR-T) cells using CRISPR/Cas9 gene editing holds huge potential to improve the efficacy and safety of T cells-based cancer therapy. The ease of use and high efficiency offered by CRISPR/Cas9 gene editing has enabled efficient gene knockout, site-specific knock-in, and genome-wide screening in T cells [19,20]. In addition, CRISPRs are applicable in several other fields, such as natural antivirals, anti-infective systems, antimicrobials, modification of the epigenetic state of DNA/RNA, and rewriting of histone epigenetic marks [21,22].

Long-term gene therapy involves the administration of a specific genetic material (i.e., DNA or RNA) via a carrier, referred to as a “delivery vector,” which facilitates the entry of the foreign genetic material into target cells [23,24]. The delivery vectors are of two types: viral vectors and non-viral vectors. The present review focuses only on viral vectors. The commonly used viral vectors are adeno-associated viruses (AAVs), adenoviruses (Ads), or lentiviruses (LVs). 

The present review highlights the key developments and challenges in the field of viral gene therapy, followed by a discussion of the possible strategies to address these challenges.

## 2. AAV Vectors

AAV-mediated gene transfer has great potential as a therapeutic approach [8]. Most of the currently developed AAV vectors are directed toward monogenic diseases, which belong to the category of rare diseases [25]. The FDA has approved the gene therapy products based on two viral vectors, which are both AAV vectors: LUXTURNA (Spark Therapeutics, Inc.) for the treatment of patients with confirmed biallelic RPE65 mutation-associated retinal dystrophy and ZOLGENSMA for the treatment of pediatric patients below two years of age having spinal muscular atrophy (SMA) with bi-allelic mutations in the survival motor neuron 1 (SMN1) gene. The use of recombinant AAV serotypes with unique tropisms to deliver cytotoxic therapy may also be considered a local antitumor therapy approach. EBV^+^ B-cells exhibit increased susceptibility to rAAV6.2 infection. Therefore, the introduction of a functional suicide gene into the rAAV6.2 genome could serve as a candidate vector for the development of rAAV-based oncolytic therapy targeting focal EBV-bearing B-lymphoproliferative disorders [26]. Intracranial interferon-beta (IFN-β) gene therapy based on the local administration of AAV vectors was reported to have successfully treated non-invasive orthotopic glioblastoma models and was also effective against migrating tumors [27]. The AAV vectors have several critical properties that could be exploited for gene delivery in cancer therapy [28].

AAV was initially discovered as a contaminant in adenovirus preparations [29]. The AAV genome comprises a single-stranded DNA approximately 4.8 kilobases (kb) in length. In addition, the AAV has a small (~25 nm) icosahedral capsid composed of three types of structural proteins, namely, VP1, VP2, and VP3 [30]. AAVs are replication-deficient parvoviruses, which have traditionally required co-infection with a helper adenovirus or herpes virus to achieve efficient infection [31]. Currently, the AAV-Helper-free system is used mostly in clinical research. On their own, AAVs are thought to be non-pathogenic and are yet to be concretely linked to any of the known human diseases. AAVs have at least 12 natural serotypes, each of which exhibits different tissue tropisms (Table 3). This is mainly because of the different affinities of these serotypes to an array of primary cell surface glycoprotein receptors and secondary receptors or coreceptors. For instance, heparan sulfate proteoglycan is thought to act as a primary receptor for AAV-2. The reported co-receptors for AAV include alpha V beta5 integrin, fibroblast growth factor receptor 1 (FGFR-1), and hepatocyte growth factor receptor (c-Met). The attachment of AAV-3 strain H relies on heparin, heparan sulfate, and FGFR-1 [32].

AAV is a non-enveloped virus that may be engineered to deliver DNA to target cells. The virus genome is not integrated into the host cell but rather forms episomal concatemers in the host cell nucleus. These head-to-tail circular concatemers remain intact in non-dividing cells, such as neurons and cardiomyocytes, and are, therefore, capable of expressing transgenes over several months [33]. AAV vectors provide a relatively stable expression in dividing cells as well. The frequency of integration events may increase if an extremely high multiplicity infection is used or if the cell is infected in the presence of an adenoviral replicase. Recently, Dhwanil A et al. indicated that chromosomal integrations occurred at a surprisingly high frequency of 1–3% both in vitro and in vivo [34]. Moreover, according to recent research, high copy numbers of the AAV9 vector led to severe toxicity in animal models [35]. 

The AAV vectors are usually preferred for in vivo gene therapy due to several advantages, including the ability to transduce both dividing and quiescent cells, robust in vivo transduction efficiency, long-term transgene expression in quiescent cells, tropism for specific tissues and cell types, relatively low immunogenicity, non-pathogenicity, and a history of clinical safety (Figure 2). 

However, certain major obstacles limit the widespread application of AAV vectors, such as: (1)Insert size: The recommended maximum insert size for cloning into AAV vector is limited. In order to counter this problem, the AAV genome is combined, via homologous recombination, to the same inverted terminal repeat (ITR) sequences. Large volumes of gene expression cassettes are divided into two or more vectors and then transported to the same cells. Two or three separate AAV vectors have been delivered successfully in animals and enabled the successful expression of functional dystrophin. The other approach involves gene fragment cutting, which is aimed at larger gene fragments. Only the functional regions are intercepted, such as delivering the B-domain-deleted factor VIII gene [36]. In addition, essentially 96% of the AAV genome may be removed to permit the engineering of the AAV vector for gene therapy [37].(2)Targeted tissue specificity: In off-target tissues, cell gene expression may cause toxicity or induce unwanted immune responses. Christos Kiourtis et al. reported that AAV8-TBG vectors serve as reliable and efficient tools for hepatocyte-specific genetic manipulation with minimal off-target effects [38]. Moreover, Reifler Aaron et al. demonstrated that compared to Opn4Cre mice, a recombinant serotype-2 adeno-associated virus (rAAV2-Opn4-Cre)-mediated Cre recombinase expression in melanopsin ganglion cells occurred without leaky expression in rod/cone photoreceptors [39]. Improving the targeted tissue specificity in gene therapy would assist in enhancing the efficacy of the therapy. The tissue specificity of AAV vector-based tissue targeting is determined by the capsid proteins of AVV. Previous studies have used the capsid and viral machinery derived from the AAV serotype 2 (AAV2), which continues to be the basis for most AAV systems, although engineered capsids, such as DJ and DJ8, which exhibit tissue-specific tropisms or higher infectivity, have become available now [40]. Different AAV serotypes exhibit different tissue tropisms and are usually applied to different clinical studies (Table 3). In addition, selecting appropriate tissue-specific promoters is important [41]. Trials have reported the use of tissue-specific strong promoters, such as albumin and synapsin, to achieve an expression specific to a particular tissue [42,43]. Systemic diseases usually require high doses of particles to be administered for clinical trials [44]. While high doses are necessary to achieve sufficient transgene expression in the target cell populations, they may lead to severe adverse effects due to off-target expression, such as hepatotoxicity [45], neurotoxicity [46], atypical hemolytic uremic syndrome (aHUS) [47], and even death in critical cases [35,48,49,50]. Increased target specificity of rAAVs would reduce the necessary viral dose as well as the off-target adverse effects. Therefore, it is imperative to develop AAV gene delivery vectors that are optimized for cell-type-specific delivery.(3)Inefficient transduction: AAV vectors may be engineered at the transgene level, for example, to optimize codons, promoters, and cis-elements, which may have the greatest potential to positively impact all AAV vectors used in the clinic [37].(4)Immune response: According to published studies, >90% of humans have been infected with AAV, while ~50% of humans may have neutralizing antibodies (Nabs) [37,51,52]. These antibodies could stimulate the production of inflammatory molecules, activate cell-death pathways, and induce the development of killer T cells capable of targeting the AAV-containing cells for destruction. Further studies revealed a set of memory CD8+ T cells against AAV capsid in humans (who have been naturally infected with AAV), which could eliminate transduced hepatocytes [53]. In addition, a considerable prevalence of neutralizing antibodies against AAV (particularly against serotype 2) has been reported in the human population, which could block the gene transfer to the liver above a certain titer. While using AAV vectors alone does not elicit a strong immune response similar to that elicited upon using other viruses such as adenovirus, the above findings highlight that the immune system remains an obstacle in the in vivo gene transfer. Anastasia Conti reported that a drug named Anakinra reduces the inflammation induced by gene editing.(5)Impractical production strategies and low viral quantities [54]: Industries and research institutions should explore how to reduce the number of plasmids required for transient transfection and improve transfection efficiency. In addition, how to increase cell culture density, expand the production capacity, and remove the empty virus should be explored.(6)High cost: The high cost of gene therapy research (such as the high cost of plasmids) and development has led to extremely high terminal commercial pricing. For instance, Zolgensma costs $2.125 million. This high cost, combined with the lack of an insurance payment system, is an issue for patients unable to afford such high costs on their own. Evidently, the traditional payment mechanisms are not adequate for gene therapies, which raises the necessity of adopting novel and efficient payment mechanisms.(7)Disease case narrow: The analysis data demonstrate that the proportion of trials for agents targeting regions other than the eye, liver, muscle, and CNS is low. Major organs, such as the heart, the kidney, and the lung, continue to be almost inaccessible to AAV-based gene therapies [55]. Several ongoing AAV gene therapy trials could translate into novel products being approved for clinical use in the future [56].

Despite certain disadvantages, AAV vectors nonetheless hold great potential to revolutionize the clinical management of human diseases.

## 3. Ad Vectors

Ad is a large and complex, non-enveloped, double-stranded DNA (dsDNA), icosahedral virus, which is 70 to 90 nm in size. Ad possesses an icosahedral protein capsid that accommodates a 26–45-kb linear, double-stranded DNA genome. Over 100 serologically different types of adenoviruses exist, among which 49 types infect humans [57,58]. According to their specific type, these viruses may bind to various cell surface proteins to facilitate their entry into the target cells [59]. As gene therapy tools, the high efficiency of Ads has resulted in over 450 protocols being approved so far for clinical trials [60].

Similar to AAV, Ad does not integrate into the host genome. Ad is the most efficient gene delivery system for a broad range of cell and tissue types. This is because most human cells express the primary adenovirus receptor and the secondary integrin receptors, such as Coxsackie and Adenovirus Receptor (CAR), CD46, and desmoglein-2 (DSG-2), as well as the glycans GD1a and polysialic acid [61,62]. Ad was the first DNA virus to enter rigorous therapeutic development, largely because of its well-defined biology, genetic stability, large transgene capacity (up to 36 kb), and ease of large-scale production. In addition, Ad leads to side effects that are considerably milder compared to chemotherapy [63,64,65]. Adenoviral vectors were initially used for brain cell transduction in the early 1990s [66]. The non-human canine adenovirus type 2 (CAV-2)-based vectors are capable of directing a gene to the neurons in the brain, spinal cord, and peripheral nervous system [67]. Adenovirus vectors may be divided into two groups: (1) replication-deficient viruses and (2) replication-competent, oncolytic viruses (OVs) [64]. The most commonly used adenoviral vector is the human Ad serotype 5, which is a common cold virus that circulates in humans with a seropositivity rate of 40–60% [68]. This virus has been rendered replication-defective through the deletion of the E1 and E3 genes [69]. The other contemporary Ad vectors have been derived from human adenovirus serotype 2 (HAd2) [70].

So far, three generations of adenoviral vectors have been developed. The first generation of Ad vectors was engineered by replacing the E1A/E1B region with transgene cassettes that could be up to 4.5 kb in length. These Ad viral vectors could induce high-level innate inflammatory responses within the first 24 h of transduction [71]. In the second generation of adenoviral vectors, the transgene capacity was enhanced further by additionally deleting the E2/E4 site, although the overall production yield remained low due to the decreased replication ability in producer cell lines [72]. The third-generation adenovirus vectors, also referred to as the helper-dependent or gutless adenovirus, have all of their viral sequences deleted, except for the ITRs and the packaging signal. The associated in vivo immune response in these viral vectors is highly reduced compared to the first- and second-generation adenovirus vectors, although high transduction efficiency and tropism are maintained [73].

Ad vectors are the most commonly used vectors in cancer gene therapy. Ad vectors are also used in vaccines to express foreign antigens [74]. Among all diseases, cancer remains the leading cause of death worldwide and accounted for nearly 10 million deaths in 2020 [75]. The overall risk accumulation is combined with the tendency for cellular repair mechanisms to become less effective as the individual grows older [76]. Cancer may be treated with surgery, radiation therapy, and/or systemic therapy (chemotherapy, hormone therapy, and targeted biological therapy). However, traditional treatments, such as surgery, may lead to side effects, including the inhibition of cellular immunity, reduction in the activity of natural killer cells, and reduction in the levels of anti-angiogenic factors [77,78,79]. Recently, viral vector gene therapy has received much attention as a novel treatment modality for cancer because of the flexibility and effectiveness it offers [80,81].

Most cases of cancer, when detected at an advanced stage, cannot be cured with traditional therapeutic modalities. Therefore, to improve tumor penetration and local amplification of the antitumor effect, oncolytic agents were developed, such as the conditionally replicating adenoviruses (CRAds). Viral infection in tumor cells results in the replication, oncolysis, and subsequent release of the virus progeny. Importantly, this replication cycle allows for a dramatic local amplification of the input dose. In theory, CRAds would replicate until all cancer cells are lysed [82]. On the other hand, similar to the other types of oncolytic virotherapy, oncolytic adenoviruses may, in addition to de-bulking the tumor, elicit powerful antiviral and antitumor immune responses. These viruses may transform a cold immunosuppressive tumor into one which is inflamed [83,84]. In other words, antitumor immunity is more important than direct oncolysis, as the former allows for the generation of tumor-specific memory T cells [65,85]. Consistent with this, the 2018 Nobel Prize in Physiology or Medicine was awarded for the discovery of cancer therapy based on the inhibition of negative immune regulation. Immune checkpoints (ICPs), in addition to controlling autoimmunity, play a key role in host defenses aimed at eradicating pathogenic microbes and microbial strategies, while also regulating the balance among tolerance, autoimmunity, infection, and immunopathology [86]. The antibodies targeting the T cell inhibitory checkpoint proteins, namely, cytotoxic T-lymphocyte-associated protein 4 (CTLA-4), programmed cell death 1 (PD1) protein, and the PD1 ligand (PDL1), have been approved for the treatment of a variety of cancers, including melanoma, non-small-cell lung cancer (NSCLC), head and neck cancer, bladder cancer, renal cell carcinoma (RCC), hepatocellular carcinoma, and several types of tumors [87].

In addition, adenoviral vectors may be used in therapeutic cancer vaccines. These vaccine adenoviral vectors are capable of inducing both innate and adaptive immune responses in mammalian hosts [88]. An example is ETBX-011, which has been developed to treat patients with cancers that express the carcinoembryonic antigen [89]. Another example would be Ad-E6E7, which generates an enhanced immune response against HPV-positive tumors [90]. In particular, great progress has been achieved recently in utilizing the Ad-based vectors as a vaccine platform for HIV and cancer immunotherapy approaches as well as in the vaccination for other infections. The recent pandemic of coronavirus disease 2019 (COVID-19), which was caused by severe acute respiratory syndrome coronavirus 2 (SARS-CoV-2), has led to an unprecedented development of multiple vaccines. Among these vaccines, Ad-vectored vaccines are also playing important roles in the global vaccine efforts against COVID-19. Certain examples include Ad26.COV2.S, ChAdOx1 nCov-19, Ad5 nCoV, and Gam-COVID-Vac vaccines, all of which have demonstrated efficacy in protecting against symptomatic COVID-19 disease in humans [91]. Despite these successes, the innate and pre-existing immunity against Ad vectors remains a serious challenge in the development and application of these vectors [92]. Moreover, according to clinical records, the administration of an adenovirus serotype 5 (Ad5) vector in a gene therapy trial led to lethal systemic inflammation in the subject [93]. One approach that could be used to overcome this obstacle is to sequentially administer two or more antigenically distinct viruses. This approach would ensure that the specific immunity that arises after the administration of the first virus does not inhibit the therapeutic effects of the second virus [94]. In addition, various non-human Ad vectors have been considered for development. Anurag Sharma et al. reported that the non-human adenovirus (Ad) vectors derived from bovine Ad serotype 3 (BAd3) or porcine Ad serotype 3 (PAd3) could circumvent the pre-existing immunity against human Ad (HAd) [95]. 

Although adenoviruses are tissue-specific and flexible, an intravenous administration of these viruses may induce acute liver injury, as has been reported in animal models [96]. In comparison to AAV, Ad has a short duration of expression in vivo [97]. Ad vectors have been studied in rodents, primates, and humans, and variable results have been achieved, which highlights the necessity for further detailed investigations on the natural history of Ad infection in humans and for questioning the value of animal models in determining the safety of virus vectors [96]. However, developing an ideal model that mimics the human infection remains the key focus of biomedical research.

## 4. LV Vectors

LVs belong to the orthoretroviridae subfamily of the genus retroviruses [98]. LVs may be divided into two major classes—primate and non-primate LVs [99]. The morphology and genome organization of all LVs are similar in several aspects: all LVs are pleomorphic spherical-shaped particles with diameters of approximately 100 nm [100], containing a diploid genome comprising two single-stranded positive-sense RNA molecules. LV vectors typically included the following required elements: 5′ long terminal repeat (LTR) through the Ψ packaging signal, central polypurine tract/chain termination sequence (cPPT/CTS), Rev responsive element (RRE), and 3′ LTR, including the poly (A) signal [101]. The classification and the specific structures of different LVs have been detailed in a previous report [9].

Currently, four generations of lentiviral vectors have been developed. The first-generation lentiviral vectors contained a significant portion of the HIV genome and exhibited a high frequency of transfer of genetic material into the host cell [102]. The lentiviral accessory genes *vif*, *vpr*, *vpu*, and *nef*, and LV regulatory genes *tat* and *rev*, were included in the first-generation LV vectors [102]. In order to achieve further safety, the second-generation LV vectors were developed by removing *vif*, *vpr*, *vpu*, and *nef*, which used to be present in the first-generation of LV vectors, as these are not necessary for the transfer of genetic material to the host cell [103]. The third-generation LV vectors are considered to be replication-incompetent and self-inactivating vectors. In this generation, the viral *tat* gene, which is essential for the replication of the wild-type human immunodeficiency virus type 1 (HIV-1), has been deleted. In addition, the vector packaging functions have been separated into three separate plasmids rather than two plasmids to reduce the risk of recombination during plasmid amplification and viral vector manufacture. An altered 3′ LTR renders the vector “self-inactivated”, which prevents the integrated genes from being repackaged. A heterologous coat protein [e.g., vesicular stomatitis virus G protein (VSV-G)] is used in place of the native HIV-1 envelope protein, and such vectors allow for the infection of a broad range of host cell types. These reasons render the third generation of LV vectors safer than the second generation LV vectors, which has allowed for the widespread application of the former [104]. The third-generation LV vectors generate virus particles using four plasmids and a producer cell line. The rationale behind including four plasmids is to enhance safety, as separating genetic components reduces the chances of recombination [105]. However, homologous recombination between the constructs remains possible nevertheless since the RRE sequence and a part of the packaging sequence in the gag gene are present in both transfer and structural packaging constructs. In order to resolve these concerns, the RRE sequences were replaced with heterologous sequences that have a similar function and do not require the REV protein. Another approach to resolving the above-stated issue is based on codon optimization. These described solutions led to the emergence of the fourth generation of LV vectors. However, the titers had been affected in the fourth-generation LV vectors, which has limited their extensive application [106].

LVs offer several potentially unique advantages over traditional gene delivery systems. Unlike adenoviral or adeno-associated vectors, neutralizing antibodies are rarely generated against lentiviral vectors [107]. The most important advantage of LV vectors is their ability to provide long-term and stable gene expression, which is crucial for adolescents or pediatric patients; LV vectors are capable of infecting dividing/non-dividing cells, such as neurons [108] and osteocytes [109], and due to their relative low-immunogenic characteristics [110], LV vectors may incorporate constructs up to 9~10 kB in size [107,111].

LV vectors are mainly used in ex vivo gene therapies (Figure 3), such as the one for B-cell acute lymphoblastic leukemia (B-ALL) [112]. B-ALL is a clonal malignant disease that originates in a single cell. B-ALL is characterized by the accumulation of blast cells that are phenotypically reminiscent of the normal stages of B-cell differentiation [113]. B-ALL remains a leading cause of non-traumatic death in children, and most adults diagnosed with it also succumb to the disease [114]. CAR-T therapy has successfully achieved extraordinary clinical outcomes in the treatment of B-ALL [115]. In order to develop CAR T-cells ex vivo, LVs appear particularly appealing due to their ability to stably integrate relatively large DNA inserts [116]. In the CAR T-cell therapy, the patient’s T-cells, either CD4+ or CD8+, are isolated and activated prior to transduction. The CAR transgene is then delivered into the activated T-cells via LV vectors and then expanded. Finally, the produced CAR T-cells are formulated in an adjusted buffer in a defined ratio of CD4^+^:CD8^+^ CAR T-cell [116,117]. A complete explanation of engineered CAR-T cells in cancer immunotherapy would not be provided in the present report and one could refer to other reports for the same [118,119,120]. The emergence of CAR-T cell therapy has paved a new way for cancer treatment. In 2017, Novartis received its first FDA approval for a CAR-T cell therapy, Kymriah (TM) (CTL019), for children and young adults with B-cell ALL that is refractory or has relapsed at least twice. In 2019, EMA approved Zynteglo, a medicine used for the treatment of patients aged 12 years and older with transfusion-dependent β-thalassemia (TDT) who do not have a β0/β0 genotype and for whom hematopoietic stem cell (HSC) transplantation is appropriate although a human leukocyte antigen (HLA)-matched related HSC donor is not available. Zynteglo (betibeglogene autotemcel) is a genetically modified autologous CD34+ cell-enriched population that contains HSCs transduced with the LV vector encoding the βA-T87Q-globin gene. LV vectors have also been demonstrated as efficient gene transfer vehicles for human solid tumor cells, such as ovarian cancer cells [121], prostate cancer [122], and hepatocellular carcinoma [123].

The major concerns associated with LV vectors-based gene therapy include the possible generation of replication-competent LVs during vector production, mobilization of the vector by endogenous retroviruses in the genomes of patients, insertional mutagenesis that may lead to cancer, germline alteration resulting in trans-generational effects, and dissemination of new viruses from the gene therapy patients [108]. LV vectors typically insert into the host DNA as a single non-rearranged copy, and while these vectors exhibit improved stability and durability, the random insertion method nonetheless has the risk of activating the cancerous gene in the genome. Several products of Bluebird that are based on lentivirus vectors have led to such events in the clinical stage. For instance, a patient was detected with myelodysplastic syndrome [124], while another patient had developed acute myeloid leukemia after treatment with LentiGlobin gene therapy [125]. In CAR T-cells therapy for ALL patients, serious although manageable adverse events, including B-cell aplasia, tumor lysis syndrome, and cytokine release syndrome, have been reported. Basically, the use of non-integrating LVs (NILVs) reduces insertional mutagenesis and the risk of malignant cell transformation due to the integration of the lentiviral vectors [126]. As stated earlier, the usage of VSV-G alters the native tropism of lentiviral vectors to allow for the infection of a broad range of host cell types [127,128,129], which implies targeting such viruses to particular cell types is challenging due to non-tissue-specificity [129,130].

However, the observed differences could have been due to the differences in vector design, final formulation, immunomodulatory regimens (transient around vector administration), and surgical approach, among other reasons. With extensive and detailed studies on LV vectors over the past few years, this platform has been used widely in both research and clinical trials. Although certain problems remain to be addressed, the safe and efficient LV vectors are nonetheless considered promising as a tool for human gene therapy.

## 5. Other Viral Vectors

Other unspecified viral vectors include vaccinia, measles, herpes simplex, alphavirus, vesicular stomatitis, influenza, baculoviruses, etc. The vaccinia virus is capable of selectively replicating and propagating productively in tumor cells, resulting in oncolysis. In addition, its rapid viral particle production, wide host range, large genome size (approximately 200 kb), and safe handling render the vaccinia virus a suitable vector for gene therapy [131,132,133]. The measles virus is a non-integrating RNA virus with a long-standing safety record in humans when used as a vector for gene therapy. The measles virus offers a novel reprogramming platform for genomic modification-free iPSCs amenable for clinical translation [134]. The herpes simplex virus (HSV) also offers numerous advantages as a vector for delivering specific genes to the nervous system, including its large size, wide host range, and its ability to establish long-lived asymptomatic infections in neuronal cells [135]. Alphavirus vectors represent an attractive approach for gene therapy applications as these offer rapid and simple recombinant virus particle production and a broad range of mammalian host cell transduction [136]. The vesicular stomatitis virus (VSV) represents an attractive oncolytic virotherapy platform owing to its potent tumor cell-killing and immune-stimulating properties. However, this vector also presents certain challenges, such as inefficient systemic delivery, which might cause severe side effects, including neurotoxicity [137]. The influenza virus is a respiratory pathogen with a negative-sense, segmented RNA genome. The construction of recombinant influenza viruses in the laboratory was first reported in the 1980s. Different gene modifications result in influenza viruses with attenuated pathogenicity, which increase the safety profile of the influenza virus vector for use in cancer gene therapy [138]. Baculovirus has been used widely for recombinant protein production in insect cells for several years. It has also been developed into safe and efficient vectors for gene delivery, which offer advantages such as broad entry tropism and replication deficiency in mammalian cells [139].

Different viral vectors offer different advantages. The different characteristics of these viral vectors allow for their application in various aspects of the gene therapy landscape. The currently approved viral vector-based gene therapy products are listed in Table 2. These viral vectors complement the therapeutics used in the treatments for diseases, thereby rendering such treatments further diverse and selective.

## 6. Conclusions

Exceptional advancements have occurred in the last few years in the biopharma sector in terms of technology, which has led to a true paradigm shift in this industry. The advent and progress of novel therapeutic approaches using virus vector-based gene therapies have undoubtedly been important milestones in this sector. Currently, products are being developed across a wide range of indications, including autoimmune, cancer, ophthalmic, neurological, alimentary/metabolic, and sensory diseases.

Viral vectors that have been artificially modified to lose their pathogenicity are used widely as delivery systems in the field of gene therapy, with the key advantages being their naturally high transduction efficiency and stable expression. However, certain challenges remain, and these challenges vary from one viral vector to another. For instance, in AAV vectors, small-scale manufacturing under cGMP is a limitation that affects the speed of development and manufacture of gene therapies. In the case of Ad vectors, deciphering how to cope with the unwanted immune responses to therapies, which leads to severe side effects, is critical. Therefore, the attempt is to develop treatments for novel conditions. In LV vectors, the focus is on developing strategies to limit the risk and improve safety [140]. The advent of CRISPR/Cas9 has shifted the direction of the field of gene therapy from gene supplementation to gene editing. As a rapid modification of the CRISPR/Cas9 system, including its delivery system, this technology has been applied extensively in preclinical and clinical treatments. However, significant challenges have to be dealt with before the CRISPR/Cas technology could be used routinely in the clinic, among which the off-target effect is one of the major concerns, and greater research should be focused on limiting its impact [141].

The clinical use of viral vector-based gene therapies has been increasing rapidly, although the development of novel clinical frameworks for adequately assessing and managing the potential delayed effects of these novel therapies must be ensured. Gene therapies, somatic cell therapies, tissue-engineering medicines, and advanced therapy medicinal products (ATMPs) together represent the four main ATMP product groups. Most gene therapies are designed to achieve permanent or long-lasting effects in the human body, which inherently increases the risk of delayed adverse events, such as the use of adenovirus vectors [142]. It is necessary to continue studying the adverse effects of gene therapy, especially long-term adverse effects. Advances in virology and an improved understanding of viral biology would contribute to the development of viral vector variants that are suitable for translational applications, while also leading to better disease treatment outcomes.

## Figures and Tables

**Figure 1 ijms-24-07736-f001:**
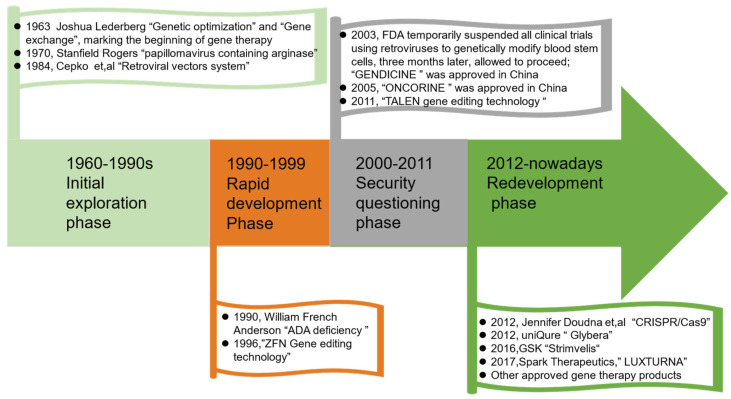
The history of gene therapy [6,7].

**Figure 2 ijms-24-07736-f002:**
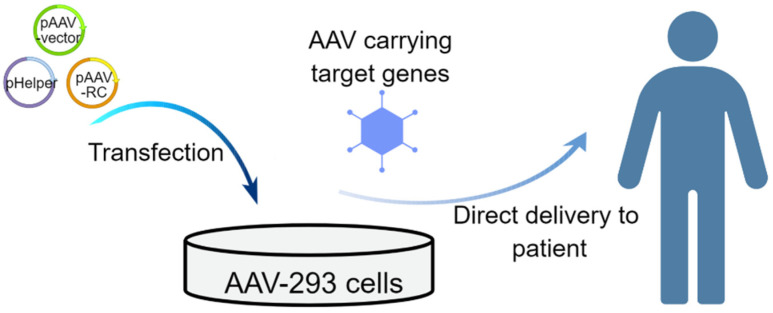
Schematic of the in vivo strategies that use AAV vectors for treating genetic diseases. The AAV Helper-Free System allows the production of infectious recombinant AAV virions without requiring the use of a helper virus. Most of the adenovirus gene products are supplied on the plasmid pHelper (i.e., E2A, E4, and VA RNA genes) that is co-transfected into AAV-293 cells. The wild-type AAV-2 genome comprises the viral rep and cap genes (for encoding replication and capsid genes, respectively), which are flanked by inverted terminal repeats (ITRs) that contain all the cis-acting elements necessary for replication and packaging. The rep and cap genes are removed from the viral vector and are supplied in "trans" on plasmid pAAV-RC. The AAV vectors have a natural ability to deliver genetic material into cells and may, therefore, be administered directly to the patient.

**Figure 3 ijms-24-07736-f003:**
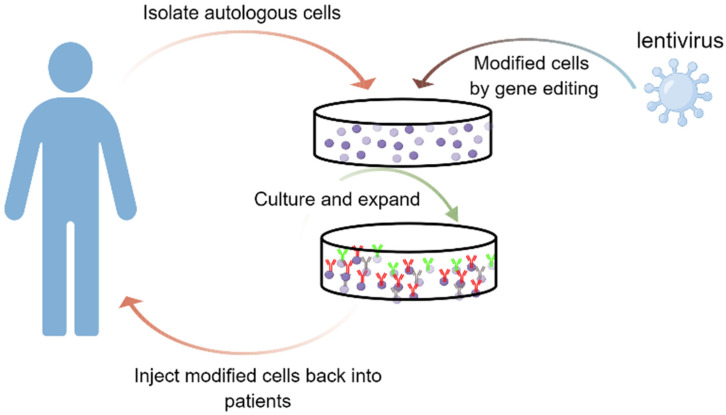
Schematic for the ex vivo strategies using LV vectors for treating genetic diseases. Cells are removed from the patient, genetically modified using LV vectors, and then returned to the patient.

**Table 1 ijms-24-07736-t001:** Key viral vector-based gene therapy products in 2022.

Drug	Development Stage	Mechanism of Action/Target Gene	Indication	Manufacture	Vector
ET140203	Phase II Clinical Trial	Immuno-oncology therapy	Hepatocellular (liver) cancer (HCC) (including secondary metastases)	Eureka Therapeutics	Lentivirus
OTOF-GT	Preclinical	Otoferlin	Hearing loss general	Sensorion	Adeno-associated virus
AVR-RD-02	Phase II Clinical Trial	Glucocerebrosidase beta	Gaucher’s disease	Avrobio	lentivirus
LX1004	Phase II Clinical Trial	CLN2	Neuronal ceroid lipofuscinosis (NCL)	Lexeo Therapeutics	Adeno-associated virus
SRP-9001	Pre-registration	Micro-dystrophin	Duchenne muscular dystrophy (DMD)	Sarepta Therapeutics	Adeno-associated virus
Vyjuvek	Pre-registration	COL7A1	Epidermolysis bullosa	Krystal Biotech	Herpes simplex virus
OCU400	Phase II Clinical Trial	NR2E3	Retinitis pigmentosa (RP) (ophthalmology)	Ocugen	Adeno-associated virus
Lumevoq	Pre-registration	ND4	Leber’s hereditary optic neuropathy (LHON)	Genethon and GenSight Biologics	Adeno-associated virus
AVB-PGRN	Preclinical	Progranulin	Dementia, frontotemporal	AviadoBio	Adeno-associated virus
ADVM-022	Phase II Clinical Trial	Aflibercept	wet age-related macular degeneration; diabetic retinopathy and other retinal conditions	Adverum Biotechnologies	Adeno-associated virus
ADVM-062	Preclinical	L-opsin	Achromatopsia	Adverum Biotechnologies	Adeno-associated virus
4D-125	Phase II Clinical Trial	Retinitis pigmentosa GTPase regulator	Retinitis pigmentosa	4D Molecular Therapeutics Roche	Adeno-associated virus
NR-082	Phase III Clinical Trial	NADH dehydrogenase subunit 4	Leber’s hereditary optic neuropathy	Wuhan Neurophth Biotechnology	Adeno-associated virus
ATA-100	Phase II Clinical Trial	Fukutin related protein	Dystrophy, limb-girdle muscular, type 2I	Atamyo Therapeutics	Adeno-associated virus
SBT101	Phase II Clinical Trial	ATP binding cassette subfamily D member 1	Adrenoleukodystrophy	SwanBio Therapeutics	Adeno-associated virus
ASC-618	Phase II Clinical Trial	B domain deleted liver-codon optimized factor VIII	Hemophilia A	ASC Therapeutics	Adeno-associated virus

**Table 2 ijms-24-07736-t002:** The currently approved viral vector-based gene therapy products.

Trade Name	Generic Name	Locations Approved	Price *	Indication	Manufacturer	Vector
ABECMA	Idecabtagene vicleucel	US, EU, JP, CAN	$419,500	Multiple myeloma.	Celgene Corporation, a Bristol Myers Squibb Company.	Lentiviral
ADSTILADRIN	Nadofaragene firadenovec	US	$158,600–$262,000	High-risk Bacillus Calmette-Guérin (BCG)—unresponsive. Non-muscle invasive bladder cancer (NMIBC) with carcinoma in situ (CIS) and with or without papillary tumors	Ferring Pharmaceuticals A/S.	Adenoviral
BREYANZI	Lisocabtagene maraleucel	US, EU, CAN, JP, UK, CH	$470,939.53	Large B-cell lymphoma (LBCL).	Juno Therapeutics, Inc., a Bristol Myers Squibb Company.	Lentiviral
CARVYKTI	Ciltacabtagene autoleucel	US, EU, UK, JP, CAN	$465,000	Relapsed or refractory multiple myeloma.	Janssen Biotech, Inc.	Lentiviral
HEMGENIX	Etranacogene dezaparvovec	US	$3,500,000	Hemophilia B (congenital Factor IX deficiency).	CSL Behring LLC.	Adeno-associated virus
IMLYGIC	Talimogene laherparepvec	US, EU, CHN, UK, AUS	$65,000	Unresectable cutaneous, subcutaneous, and nodal lesions in patients with recurrent melanoma after the initial surgery.	BioVex Inc., a wholly owned subsidiary of Amgen, Inc.	Herpes simplex virus –1
KYMRIAH	Tisagenlecleucel	US, EU, UK, CAN, JP, AUS, KR, CH	$475,000	Relapsed or refractory follicular lymphoma.	Novartis Pharmaceuticals Corporation.	Lentiviral
LUXTURNA	Voretigene neparvovec	US, EU, UK, AUS, CAN, KR	$850,000	Confirmed biallelic RPE65 mutation-associated retinal dystrophy	Spark Therapeutics, Inc.	Adeno-associated virus
SKYSONA	Elivaldogene autotemcel	US, EU, UK ^#^	$3,000,000	Neurological dysfunction in boys aged 4–17 years with early, active cerebral adrenoleukodystrophy (CALD).	Bluebird Bio, Inc.	Lentiviral
TECARTUS	Brexucabtagene autoleucel	US, EU, UK, CAN	$373,000	Relapsed or refractory mantle cell lymphoma (MCL) and B-cell precursor acute lymphoblastic leukemia (ALL).	Kite Pharma, Inc.	Retroviral
YESCARTA	Axicabtagene ciloleucel	US, EU, UK, JP, CHN, CAN	$373,000	Large B-cell lymphoma.	Kite Pharma Inc.	Retroviral
ZYNTEGLO	Betibeglogene autotemcel	US, CAN, EU ^#^	$2,800,000	Adult and pediatric patients with ß-thalassemia who require regular red blood cell (RBC) transfusions.	Bluebird bio Inc.	Lentiviral
ZOLGENSMA	Onasemnogene abeparvovec	US, EU, JP, AUS, CAN, BRA, TWN, KR, ISL, NO, LIE, UK	$2,125,000	Spinal muscular atrophy (Type I)	Novartis Gene Therapies, Inc.	Adeno-associated virus
GLYBERA	Alipogene tiparvovec	EU ^#^	$1,000,000	Lipoprotein lipase deficiency.	uniQure biopharma.	Adeno-associated virus
LIBMELDY	Atidarsagene autotemcel	EU, UK, ISL, NO	$3,780,000	Metachromatic leukodystrophy (MLD).	Orchard Therapeutics (Netherlands).	Lentiviral
ROCTAVIAN	Valoctocogene roxaparvovec	EU, UK	Around $2,500,000	Hemophilia A (congenital factor VIII [FVIII] deficiency).	BioMarin International Limited.	Adeno associated virus
STRIMVELIS	autologous CD34+ enriched cells	EU, UK	$665,000	Severe combined immunodeficiency due to adenosine deaminase deficiency (ADA-SCID).	Orchard Therapeutics (Netherlands).	Retroviral
UPSTAZA	Eladocagene exuparvovec	EU, UK	more than $3,600,000	Severe deficiency of aromatic L-amino acid decarboxylase (AADC).	PTC Therapeutics International Limited.	adeno associated virus
GENDICINE	Recombinant p53 gene	CHN	N/A	Head and neck cancer.	Shenzhen SiBiono GeneTech Co. Ltd.	Adenoviral
ONCORINE	E1B/E3 deficient adenovirus	CHN	N/A	Head and neck cancer; Nasopharyngeal cancer.	Shanghai Sunway Biotech Co. Ltd.	Adenoviral
REXIN	G mutant cyclin-G1 gene	PHL	N/A	Solid tumors.	Epeius Biotechnologies.	Retroviral
DELYTACT	Teserpaturev	JP	$12,500	Malignant glioma.	Daiichi Sankyo.	Herpes simplex virus –1
RELMA-CEL	Relmacabtagene autoleucel	CHN	N/A	Diffuse large B-cell lymphoma	JW Therapeutics.	Lentiviral
ZALMOXIS	N/A	EU ^#^	$263,000	Haploidentical hematopoietic stem cell transplantation (HSCT) of adult patients at a high risk of hematological malignancies.	MolMed S.p.A.	Retroviral
INVOSSA-K	N/A	KR ^#^	N/A	Osteoarthritis.	Kolon Life Science.	Retroviral
RIGVIR	N/A	LV, EE, PL, AM, BLR	N/A	Local treatment of skin and subcutaneous metastases of melanoma.	SIA Latima.	Echovirus 7

Data statistics as of 2023/01. ^#^: Withdrawal of the marketing authorization. *: Pricing varies depending on the country. N/A: Not available.

**Table 3 ijms-24-07736-t003:** The tissue tropisms of different AAV serotypes and the representative clinical trials.

AAV Serotype	Tissue-Specific Tropisms	Key Pipeline	Disease	The Delivered Gene	Sponsor (s)	The Clinical Trial Stage
AAV1	Muscle, heart, skeletal muscle (including cardiac muscle), nerve tissue	Glybera	Lipoprotein lipase deficiency	Lipoprotein lipase	UniQure	Approved ^#^
AAV2	Central nervous system, muscle, liver, brain tissue, eye	BIIB111	Hereditary ophthalmopathy	Rab escort protein 1	NightstaRx Ltd., a Biogen Company	Phase III completed, suspended
AAV3	Muscles, liver, lung, eye	N/A	N/A	N/A	N/A	N/A
AAV4	Central nervous system, muscle, eye, brain	N/A	N/A	N/A	N/A	N/A
AAV5	Lung, eye, central nerve, joint synovium, pancreas	BMN 270	Hemophilia type-A	Coagulation factor VIII	BioMarin Pharmaceutical	Approved
AAV6	Lung, heart	SB-525	Hemophilia type-A	Coagulation factor VIII	Pfizer	Phase III
AAV7	Muscle, liver	N/A	N/A	N/A	N/A	N/A
AAV8	Liver, eye, central nerve, muscle	BIIB112	X-linked retinoschisis	Pigmentosa GTPase regulator	NightstaRx Ltd., a Biogen Company	Phase III, suspended
AAV9	Heart, muscle, lung(alveolar), liver, central nervous system	PF-06939926	Duchenne muscular dystrophy	Truncated dystrophin	Pfizer	Phase III
AAV-DJ	Liver, retina, lung, kidney	N/A	N/A	N/A	N/A	N/A
AAV-DJ/8	Liver, eye, central nervous system, muscle	N/A	N/A	N/A	N/A	N/A
AAV-Rh10	Lung, heart, muscle, central nervous system, liver	LYS-GM101	GM1 gangliosidosis	beta-galactosidase	Lysogene	Phase II
AAV11	Unknown	N/A	N/A	N/A	N/A	N/A
AAV12	Nasal	N/A	N/A	N/A	N/A	N/A
AAV13	Unknown	N/A	N/A	N/A	N/A	N/A

N/A: Not available. ^#^: Withdrawal of the marketing authorization.

## Data Availability

Not applicable.

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
