# Peer review of "Viral Vector-Based Gene Therapy"

_ijms, 2023, doi:10.3390/ijms24097736_

Round 1
Reviewer 1 Report
I consider that it is a useful and well-developed manuscript, but it is convenient to improve several general aspects:
The paragraph on sheet 4 is very long, please divide it into several paragraphs.
I consider that each virus section is subdivided into sections, for example: Biological characteristics, history, tropism, background in gene therapy, current use, adverse effects, etc. And all the same sections for all viruses
In the adenovirus section, it would be interesting to mention in a special way the adenovirus-based vaccines against COVID, which, although they are not regulatory classified as gene therapy, biologically and technically it is the same generation strategy.
It is worth a section on ethical and regulatory aspects, as well as another section on adverse effects, mentioning that the long-term effects may still be uncertain. For example, with the use of adenovirus it has been shown that they can have long-term effects "Epidemiological studies are needed to evaluate the effect of adenovirus as a precursor of chronic liver and cardiovascular diseases, including the chronic effects of gene therapy", as mentioned in the following manuscript, which should be cited: PMID: 30666458 DOI: 10.1007/s00705-018-04132-6
A table where the vectors, countries, cost and usefulness of the therapies already commercially approved to date are mentioned would be interesting. Although this table may expire soon, it would specify what date this table is, specifying that the data may vary depending on the query date.
Reviewer 2 Report
The review is about the use of viral vectors for gene therapy. Along with that, the authors mentions the use of vectors for gene editing, oncolytic therapy and vaccination.
The major issues include:
While the authors mention gene editing in the introduction, they do not describe it nor current clinical trials and perspectives drugs in development for gene editing in the main body of the manuscript
On the contrary, they do not mention in the introduction the use of vectors for oncolytic therapy and vaccine development, but these very important and large topics are mentioned just in several sentences in the main body of the manuscript. These should represent separate chapters
Figure 1 would benefit from a graph representing the number of research studies published and clinical trials started during the phases that the authors outline
Figure 1 lacks any mentioning of RNA/DNA vaccines which in fact represent a sort of gene therapy
Figures, figure legends and the body of the manuscript requires extensive English editing
Numerous typos and misspellings occur throughout the manuscript text and the Figures
Many words are used improperly and, sometimes, shift to publicistic and popylar language style which is inappropriate in a scientific review. In particular, such wordings as “tortuous history” etc.
The “birth of CRISPR…” is not appropriate; first, because the review should be very precise in terms. Second, because CRISPR were discovered and characterized many decades ago; J. Doudna and E. Charpentier adjusted it for use in gene editing and were one of the first who proposed this idea
Along with ZFNs, TALENs and CRISPRs, meganucleases are also on par with the former two gene editing systems
Along with knockout, knock-in and screens, CRISPRs can also be used for many other applications (PMC6929090; 28581505; 28581505)
The authors state that “long-term gene therapy involves administration of … DNA or RNA. Please, specify how RNA could be used in long-term gene therapy
There are other vectors which are used widely or for specific applications, including herpesviruses, baculoviruses etc.
Table 1 lacks genes that are transferred by AAV for treating a specific disease as well as the stage of the clinical trial
The chapter of AAVs lacks mentioning severe adverse effects that AAVs induce, such as hepatotoxicity, neurotoxicity, nephrotoxicity etc. The problem of pre-existing immunity is not covered
The efficiency of AAV transduction and off-site transduction in animals and humans is missing
It is not common to see measurement of viral size in A. It would be more convenient to mention the size of virions in nm. Moreover, for consistency, the size of other vectors in addition to AdV should be mentioned
In the description of AdV, the authors mention cancer therapy, meaning oncolytic therapy. If they mention this, they should make a separate chapter, as indeed, oncolytics are very promising therapies for numerous types of cancers. Other oncolytics should be mentioned and briefly described
Round 2
Reviewer 1 Report
The manuscript was considerably improved and recommendations have been included. There are only two small aspects left where you can be more precise.
1.- In figure 1, I consider that mentioning the year in which the first gene therapy for commercial use was approved would be relevant, which I believe was in China in 2003 (Ad with p53) and another in 2005 (oncorine), also in China. These events were very relevant. I leave it to your consideration whether or not to include it in the figure, but these events have to be mentioned.
2.- Not all the long-term effects caused by gene therapy, especially the long-term adverse effects, are not necessarily produced by wanting to produce long-term beneficial effects. A short-term beneficial effect may be sought, but long-term adverse effects may remain. Please rewrite this sentence: "Most gene therapies, such as the use of adenovirus vectors, are designed to achieve permanent or long lasting effects in the human body, which and this inherently increases the risk of delayed adverse events [140]" . Mention that it is necessary to continue studying adverse effects of gene therapy, especially long-term adverse effects. The reference 140 is consistent and appropriate for this sentence.
Reviewer 2 Report
The issues were addressed, the manuscript is now better fit for publication.
Author Response
thanks for your comments !